# Toward Greater Legitimacy: Online Accountability Practices of Ukrainian Nonprofits

Asya Cooley 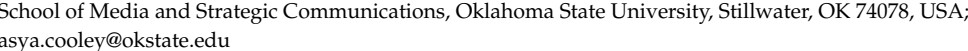

School of Media and Strategic Communications, Oklahoma State University, Stillwater, OK 74078, USA; asya.cooley@okstate.edu

**Abstract:** This exploratory study examined recent innovations in online accountability practices adopted by Ukrainian nonprofits. A quantitative content analysis of Ukrainian nonprofit websites evaluated their adoption of online accountability practices across five dimensions: accessibility, engagement, performance, governance, and mission. The results reveal wide variability in accountability scores and low average disclosure of performance and governance practices. Nonprofits without a listed location have lower scores. International NGOs demonstrate stronger governance than domestic groups. Associations are leaders in online innovations. Overall, Ukrainian nonprofits are not fully capitalizing on digital platforms to enhance transparency and stakeholder engagement. This study provides unique insights into online accountability among Ukrainian nonprofits and highlights opportunities for nonprofits to leverage websites as strategic management tools for greater accountability, legitimacy, and impact.

**Keywords:** online accountability; Ukraine; nonprofit sector; nonprofit management

## 1. Introduction

Accountability is central to nonprofit work, involving organizations explaining, justifying, and taking responsibility for their actions (Cooper and Owen 2007). Stakeholders expect nonprofits to work for the public good while meeting expectations around financial reporting, performance, governance, and more. However, researchers note that accountability remains an underexplored area, with a need to develop more knowledge on voluntary disclosure practices (Ortega-Rodríguez et al. 2020). The emphasis on transparency in the UN Sustainable Development Goals also demonstrates the significance of accountability for credibility, and the adoption of online accountability tools represents an important innovation in nonprofit management processes.

Current scholarship on accountability practices focuses predominantly on Western contexts. This study examined the Ukrainian nonprofit sector to provide a non-Western perspective. Ukraine presents an excellent case for understanding varying nonprofit legitimacy globally. Recent crises like COVID-19 and war have impacted Ukraine's nonprofits, underscoring the need for accountability. Over USD 1.7 billion in aid has flowed to Ukrainian nonprofits since 2022, indicating massive new resources requiring oversight (Philanthropic Response to the War in Ukraine n.d.). One aspect of overall organizational accountability practices is online accountability, which refers to how an organization justifies actions and engages stakeholders through Internet technologies (Dumont 2010). Little prior research has examined nonprofit accountability in Ukraine specifically. Some scholars note that the informal, decentralized nature of Ukrainian nonprofits contributes to limited transparency, monitoring, or formal accountability mechanisms (Oleinik 2018; Shostko 2020).

This exploratory research project focused on (1) investigating the status of online accountability practices within the Ukrainian nonprofit landscape, and (2) identifying organizational factors associated with greater adoption of these innovative tools.

This project makes several key contributions to research on innovations in nonprofit management. First, we build on the legitimacy theory and explore the factors that contribute

to higher levels of nonprofit accountability. Second, we build on the available assessment tools, such as the Non-profit Virtual Accountability Index (NPVAI) (Dumont 2013), Online Accountability Practices (Cooley 2020), and navigation features from the most current website assessment tool (Al-Qallaf and Ridha 2019) to build an updated index evaluating these online innovations. Third, we contribute to the existing research landscape by addressing nonprofit accountability in non-Western contexts, as the current scholarly discourse predominantly centers around Western settings. From a practical standpoint, we intend to furnish Ukrainian nonprofits with a robust framework of accountability practices that can be readily incorporated into operational strategies, thereby enhancing transparency, stakeholder trust, and overall organizational effectiveness. The findings will help Ukrainian nonprofits understand how online innovations can strengthen transparency, legitimacy, and trust.

## 2. Literature Review

Our literature review will first examine the theoretical framework of legitimacy theory, and then explore online accountability including key concepts and definitions, followed by an analysis of the nonprofit sector in Ukraine covering the landscape overview, public trust challenges, prior research on accountability, and factors influencing online accountability.

### 2.1. Theoretical Framework: Legitimacy Theory

The legitimacy theory provides the theoretical foundation for this research project. According to this theory, nonprofit organizations aim to align their actions and behaviors with broader societal norms, expectations, and values to gain and maintain legitimacy among stakeholders and the public (Suchman 1995). Organizations seek to demonstrate, through their activities, communications, and disclosures, that they are serving socially desirable goals in ways that are acceptable and appropriate within their cultural context.

A key tenet of legitimacy theory is that an organization's perceived legitimacy significantly impacts its ability to attract resources, maintain public support, and ultimately achieve its mission. One important pillar of legitimacy emphasizes accountability—providing transparent information about an organization's activities, finances, management, and impact to assure stakeholders and the public of its legitimacy (Conway et al. 2015; Leardini et al. 2019; Ebrahim 2009).

Past scholarship has shown how accountability practices allow nonprofits to actively pursue legitimacy. For example, the 1992 high-profile scandal at United Way led many US nonprofits to increase their transparency, disclosure, and self-regulation to rebuild legitimacy in the eyes of the public and regulators (Bothwell 2004). Ongoing regional research finds variations in how nonprofits leverage different dimensions of accountability to signal legitimacy based on factors like organization type, size, location, donor base, and external environment (Dhanani 2009; Tremblay-Boire and Prakash 2015).

Utilizing new information technologies, online accountability has emerged as a key approach through which nonprofits demonstrate legitimacy to digital audiences. This involves proactively explaining and justifying organizational actions to stakeholders through detailed, two-way communication on the organization's website and social media platforms (Dumont 2010).

Drawing from legitimacy theory, this study identified and tested specific organizational and environmental factors that may drive the adoption of online accountability practices among nonprofits seeking to establish legitimacy. The theory suggests factors like nonprofit type, location, revenue sources, stakeholder expectations, and external environment may shape an organization's use of online tools to signal legitimacy. Examining these potential factors will provide theoretical and practical insights into nonprofit adoption of online accountability innovations.

*2.2. Online Accountability*

2.2.1. Definition and Key Concepts

Online accountability refers to how an organization justifies and accounts for its actions through Internet technologies and two-way communication with stakeholders (Dumont 2010). While organizational accountability has been studied extensively, accountability specifically in online spaces is less understood. The Internet provides opportunities for nonprofits to share information and engage stakeholders through websites, social media, and other innovative platforms (Jeacle and Carter 2014; Cho et al. 2009; Lowe et al. 2012; Mackenzie et al. 2013). Thus, online accountability has become an important part of nonprofits' accountability efforts and strategic management.

2.2.2. Instruments Measuring Online Accountability

Researchers have developed innovative frameworks to assess online accountability. Saxton and Guo proposed a model with two key dimensions—disclosure and dialogue—and tested it on community foundation websites. They found websites were more successful at disclosing information than enabling stakeholder engagement (Saxton and Guo 2011). Gandía analyzed Spanish nonprofits' disclosure on websites using a Disclosure Index. The study found that websites should provide more substantive information on work, funding, governance, and operations to aid nonprofit management (Gandía 2011).

The Nonprofit Virtual Accountability Index (NPVAI) is a comprehensive, validated tool measuring website accountability across several dimensions. First developed by Dumont (2013) and then tested within a hospital context by Cooley (2020), the NPVAI allows a comparative assessment of how well organizations facilitate online accountability. This index measures the extent to which nonprofit websites meet accountability expectations across five dimensions: (a) accessibility—ease of navigating the website; (b) engagement—ease of connecting with the organization; (c) performance—sharing financial/nonfinancial results and reports; (d) governance—information on leadership, bylaws, and meetings; and (e) mission—statements of mission, values, and goals.

2.2.3. Research on Nonprofit Online Accountability Practices

Despite the potential benefits, many nonprofits have not fully utilized websites and social media for transparency and accountability in their strategic management (Lee and Blouin 2019; Stevens et al. 2018). However, research shows that accountability practices on websites can increase donations. For example, disclosing IRS Form 990s correlates with higher donations, indicating usefulness for donor decision-making (Blouin et al. 2018).

During the COVID-19 pandemic, studies using the NPVAI found nonprofits focused more on attracting donors through accessibility and engagement features than sharing performance information (Uygur and Napier 2023). An analysis of social media usage for accountability by Indonesian nonprofits showed that platforms like Facebook and Instagram enabled various formal and informal accountability demands (Amelia and Dewi 2021). However, social media was used for short-term engagement rather than planned, strategic accountability and management.

Overall, the research indicates nonprofits are not fully capitalizing on websites and social media for transparency, interactivity, and accountability. More strategic use of online platforms could strengthen stakeholder trust and support through improved nonprofit management. Frameworks like the NPVAI guide comprehensive online accountability practices across multiple dimensions. We now turn to reviewing the nonprofit landscape in Ukraine, and how scholars understand Ukrainian nonprofit accountability.

*2.3. Nonprofit Sector in Ukraine*

2.3.1. Overview of the Nonprofit Landscape in Ukraine

The Ukrainian nonprofit sector has distinct characteristics. It is often described as weak (Martin and Zarembo 2023; Gatskova and Gatskov 2016) with a "notoriously low level of civic engagement" (Gatskova and Gatskov 2016). In addition, the political and

legislative environment where Ukrainian nonprofits operate is highly unstable and, at times, even antagonistic (Krasynska 2015). The Ukrainian understanding of civil society tends to emphasize informal action, values, responsibility, and a sense of community rather than just membership in formal organizations (Martin and Zarembo 2023). Not surprisingly, events like Euromaidan and the current Russian invasion have triggered massive grassroots civic mobilization (Martin and Zarembo 2023). The concept of a "sense of community" helps explain the dormant yet powerful nature of Ukrainian civil society during crises. The strong values, emotional connections, and responsibility Ukrainians feel toward their nation and community drive heightened civic activism when threats emerge (Martin and Zarembo 2023).

Notably, researchers report very low levels of public confidence and trust in the country's nonprofits (Krasynska 2015). Less than five percent of Ukrainians "fully trust" nonprofits, whereas 18 percent "fully mistrust" them (Razumkov Center 2013). Trust is a crucial precondition in people's engagement and participation in collective action (Gatskova and Gatskov 2016; Kuts and Palyvoda 2006; Sønderskov 2011), such as nonprofit work. Robust organizational *accountability* practices can be an answer to this challenge. Previous studies provide empirical evidence that accountability activities influence public attitudes toward nonprofits and contribute to financial and ethical integrity (Becker 2018). Virtually no academic literature is available on the subject of Ukrainian nonprofit accountability, and this project attempts to fill this gap.

### 2.3.2. Low Public Trust and Challenges with Legitimacy

The nonprofit sector in Ukraine faces challenges in establishing a strong sense of legitimacy within the perception of the Ukrainian public. Some academics label it as a "shadow third sector" (Krasynska 2015), whereas others highlight the challenge of low public trust, attributed by several scholars to lingering aspects of *Homo Sovieticus*—a distinct sociocultural personality marked by deindividualization (Levada 1993; Gatskova and Gatskov 2016). Heavily influenced by communist ideology, the ideal Soviet citizen was expected to be a member of several organizations (such as the communist party, trade unions, military associations, etc.), and these memberships were based on "obligation, obedience, and external conformity", rather than self-driven "internal and voluntary initiatives" (Howard 2003, p. 27). After the fall of the Soviet Union, the post-communist transformation was marked by people's unwillingness to join and maintain membership in organizations, as a backlash to years of "over-organization" of communist societies (Gatskova and Gatskov 2016, p. 680). Thus, organizational membership in Ukraine dropped from 81% to 30% in the first decade after the USSR collapse (Weßels 2003).

### 2.3.3. Prior Research on Ukrainian Nonprofit Accountability

To the best of our knowledge, there is very little previous academic research specifically examining nonprofit accountability in Ukraine. While examining Ukrainian volunteerism, one study suggested that the informal and decentralized nature of Ukrainian volunteer initiatives means limited transparency, performance monitoring, or formal accountability mechanisms (Oleinik 2018). Another scholar suggested that the decentralized nature of the nonprofit sector in Ukraine contributes to the lack of oversight and, hence, much lower levels of accountability, compared to the public sector (Shostko 2020).

Since very little literature on Ukrainian nonprofit accountability could be identified, we instead relied on and extrapolated from the available research on accountability in Ukraine's public sector. One research project tells us that the Euromaidan protest movement of 2013–2014 led to increased activism around transparency and accountability reforms in Ukraine, with civil society groups playing a major role in pushing anti-corruption initiatives (Tregub 2019). This movement contributed to gains like the creation of independent anti-corruption agencies, online procurement systems, and requirements for officials to disclose assets (Tregub 2019; USAID 2021). For example, groups like Transparency International Ukraine introduced tools to evaluate and provide feedback on local government transparency,

including a Code of Transparent Government promoting accountability best practices (USAID 2021). These online platforms allow Ukrainian residents to assess local transparency and accountability levels and communicate with officials (USAID 2021). The agency further argues for the need for continuous efforts to institutionalize public accountability mechanisms. This project contributes to the study of accountability practices, and the first two research questions address the current state of online accountability adoption among Ukrainian nonprofits and how these practices can be improved.

*RQ1: What is the current state of online accountability adoption among Ukrainian nonprofits?*

*RQ2: How can Ukrainian nonprofits enhance their online accountability? What are the major areas for improvement?*

2.3.4. Factors Influencing Online Accountability

Previous research has examined various factors that can influence the online accountability practices of nonprofit organizations. These include organizational characteristics like nonprofit type, service area, location, annual revenue, and age. For example, larger and more established nonprofits tend to have more robust accountability practices on their websites, likely due to their greater resources and capabilities (Saxton and Guo 2011). Nonprofits in service domains like education and health have also exhibited higher accountability practices online, while arts and culture nonprofits scored lower (Hsu et al. 2017). Geography can also be a factor, as urban nonprofits sometimes have more advanced online accountability than rural nonprofits (Kang and Norton 2004). In summary, a nonprofit's resources, capabilities, stakeholder expectations, and field of work all seem to influence the level of accountability present on its websites and online presence. Our third research question was set to determine these factors:

*RQ3: Which organizational factors are associated with higher levels of online accountability practices among Ukrainian nonprofits?*

Several studies have examined differences in accountability practices between domestic nonprofits and international NGOs. International NGOs demonstrate higher online transparency compared to domestic nonprofits, likely reflecting stricter accountability demands from global donors and supporters (Gandía 2011). As a result, international NGOs rely heavily on foreign donor funding for operations and prioritize upward accountability to donors over local needs (Wong 2010). This accountability to donors can lead NGOs to propose donor-driven solutions rather than locally appropriate ones (Wong 2010).

Specific examples illustrate issues with donor-driven priorities undermining nonprofit accountability. A study on US aid to women's NGOs in Ukraine found a disconnect between donor goals and local organizations' activities (Pishchikova 2010). The author argues the decentralized Ukrainian nonprofit sector appears susceptible to donor priorities that may hinder accountable outcomes, concluding that external aid can undermine domestic nonprofit accountability (Pishchikova 2010). Another study supports this, finding that Western-funded Ukrainian NGOs were criticized as unaccountable to local constituencies, prioritizing foreign donors over local needs (Gutnik 2007). While donors prioritize quantitative metrics and short-term results, this focus alone does not fully capture long-term impacts (Gutnik 2007). Given these findings, we examined any possible differences in how domestically operated nonprofits and NGOs approach online accountability practices.

*RQ4: How do online accountability practices differ between domestic Ukrainian nonprofits and international non-governmental organizations (NGOs) operating in Ukraine?*

We looked into the literature that focused on the service area of Ukrainian nonprofits and any relationships with accountability practices. One study focused on humanitarian-focused nonprofits that served Ukrainian populations after Russia invaded Ukraine in 2022

and coined the term "distributed humanitarianism", which refers to flexible, temporary aid chains rather than centralized bureaucracies (Cullen Dunn and Kaliszewska 2023, p. 19). According to scholars, these loosely organized groups delivered more tailored, responsive aid compared to slow, inflexible international agencies focused on donor accountability and bulk logistics (Cullen Dunn and Kaliszewska 2023). These findings promote further questions on service area–accountability relationships, and our final research question emerges.

*RQ5: How do online accountability practices vary based on the service area (child welfare, education, humanitarian, etc.) of Ukrainian nonprofits?*

### 3. Methods

We used the quantitative content analysis method to assess online accountability practices in Ukrainian nonprofit organizations. This method offers a systematic and objective approach to analyzing textual data, enabling us to uncover patterns, trends, and relationships with a focus on numerical representations and statistical validation.

### 3.1. Sampling

We used a purposive sampling technique since probabilistic sampling was not a feasible option due to a lack of an updated, comprehensive list of Ukrainian nonprofit organizations. While the non-random nature of a purposive sample makes assessing representativeness and generalizability difficult compared to a randomized sample, it was the best option to use for this study. Using the platforms GlobalGiving.org and Dobro.ua, as well as the search engine Google.com, we selected a total of 101 nonprofit organizations for analysis. An organization was included if it had a working and accessible website and was operating in Ukraine. Ten organizations did not match these criteria and were, therefore, excluded from the final count. A total of 91 organizations were examined. An organizational website was a unit of analysis for this study. All websites were coded in March, April, and May 2023.

### 3.2. Coding Scheme

The coding scheme consisted of two blocks. First, we recorded organizational variables: name, website, type of a nonprofit (domestic NPO, NGO, or other/unable to determine), service area (child welfare, peace, and reconciliation, education, and humanitarian), location (Kyiv, regional, international, or other/unable to determine), and age (through year established).

Next, we refined the online accountability practices (OAP) instrument that was heavily based on Dumont's (2013) instrument, and later adapted by Cooley (2020). The OAP instrument includes five measures: accessibility, engagement, performance, governance, and mission (See Appendix A for a full instrument description).

The accessibility measure was heavily revised. A revision was much needed because accessibility measure questions in the original instrument were over a decade old, and website accessibility features have evolved significantly since then. The newly updated accessibility measure was based on Al-Qallaf and Ridha's (2019) best practices and included website navigation, a search feature, quick links, a help feature, a site map, and language options. Appendix A includes the full instrument description and indicates which questions were added from the Al-Qallaf and Ridha (2019) accessibility instrument. The engagement measure examined variables like the recency of updates, newsletters/community updates, links to foundations/giving, the number of social media links, instant connectivity, and sharing options. The performance measure looked at the sharing of information like annual reports, financial statements, and mentions of accreditations/honors/awards. The governance measure covered the disclosure of organizational bylaws, the board of directors/leadership team, and board meeting minutes/summaries. Finally, the mission measure included variables like goals/strategic plans/implementation plans, employee directory, mission statement, and statement of values.

The overall OAP score for each nonprofit was calculated based on the five OAP dimensions. Each dimension was weighted on a 20-point scale, which prevented any one dimension from having an outsized influence on the total score.

### 3.3. Internal Validity and Reliability

One coder was a Ukrainian speaker who verified and checked all the codes. Reliability was ensured by employing consistent data collection protocols (such as a clearly defined instrument and thorough coder training) and using inter-coder agreement measures to demonstrate the consistency and accuracy of the obtained results. After two rounds of pilot testing 12 websites, a sufficient overall intercoder reliability was reached (using Cohen's Kappa), with an overall intercoder reliability score of 0.97 and no individual categories below 0.82.

Next, we checked for the internal (reliability) consistency of the OAP instrument. The Cronbach's alpha for the OAP scores was 0.73, which indicates good reliability for the instrument according to previously established guidelines (Field 2009).

### 3.4. Analyses

We first ran a descriptive statistical analysis of the coded variables and then used a variety of statistical tests to answer the first two research questions. To answer RQ3, a multiple regression statistical model was employed. To answer RQ4, an independent group *t*-test was used to determine overall differences between domestic NPOs and NGOs in terms of OA practices and the five individual dimensions. For the last research question, RQ5, we ran a one-way ANOVA with a Tukey post hoc test. SPSS version 26 was used for all data analyses.

## 4. Results

*RQ1: What is the current state of online accountability adoption among Ukrainian nonprofits?*

*RQ2: How can Ukrainian nonprofits enhance their online accountability? What are the major areas for improvement?*

Ukrainian nonprofit websites in our sample varied dramatically in terms of online accountability, with some websites scoring as low as 5.6 to as high as 77 on the 100-point weighted scale of the OAP. Looking at a comparative analysis across all five dimensions, Ukrainian nonprofits overall scored very low on governance ($M = 3.4$) and performance ($M = 5.9$) dimensions while reporting their online accountability. The engagement dimension was the most robust ($M = 12.3$). Accessibility scores varied the least amongst the sample ($SD = 3.2$), while performance scores varied the most ($SD = 5.7$). Notably, some websites had no information at all on performance, governance, and mission, and were considered extremely inaccessible. See Table 1 below for more details on the range of OAP dimension scores for the Ukrainian nonprofit websites analyzed.

**Table 1.** Descriptions of weighted OPA scores and individual dimensions.

|  | Weighted Min | Weighted Max | Weighted Mean | Weighted SD |
|---|---|---|---|---|
| OAP | 5.6 | 77.2 | 36.0 | 15.7 |
| Accessibility | 0 | 15.6 | 7.9 | 3.2 |
| Engagement | 2.2 | 20 | 12.3 | 4.1 |
| Performance | 0 | 20 | 5.9 | 5.7 |
| Governance | 0 | 13.3 | 3.4 | 4.2 |
| Mission | 0 | 20 | 6.5 | 5.2 |

*RQ3: Which organizational factors are associated with higher levels of online accountability among Ukrainian nonprofits?*

A multiple regression statistical model was specified as follows:

*OAP = b0 + b1 domestic (dummy) + b2 NGO (dummy) + b3 child welfare (dummy) + b4 peace and reconciliation (dummy) + b6 humanitarian (dummy) + b7 military assistance (dummy) + b8 association (dummy) + b9 Kyiv (dummy) + b10 unknown location (dummy) + b11 international (dummy) + b10 age + e*

The results of the regression indicated that the constructed model predicted 24.6% of the variation in the Online Accountability Practices scores [$F(1,64) = 2.084$, $p = 0.039$] (see Table 2). The unknown location of nonprofits was a significant predictor of lower OAP scores, while associations were generally predicted to have higher OAP scores.

**Table 2.** Summary of regression analysis results for variables predicting weighted OAP scores.

| | Unstandardized Coefficients | | Standardized Coefficients | t | Sig. |
|---|---|---|---|---|---|
| | **B** | **Std. Error** | **Beta** | | |
| (Constant) | 26.99 | 9.78 | | 2.76 | 0.01 |
| Domestic NGO (dummy) | 6.74 | 3.76 | 0.20 | 1.79 | 0.08 |
| Age | 0.00 | 0.11 | 0.00 | −0.03 | 0.98 |
| Location = Kyiv (dummy) | −0.98 | 4.11 | −0.03 | −0.24 | 0.81 |
| Location = International (dummy) | 1.80 | 5.96 | 0.04 | 0.30 | 0.76 |
| Location = NA (dummy) | −17.32 | 6.53 | −0.34 | −2.65 | 0.01 * |
| ServiceArea = Child welfare (dummy) | 14.74 | 9.12 | 0.46 | 1.62 | 0.11 |
| ServiceArea = Peace and reconciliation (dummy) | 5.15 | 13.90 | 0.05 | 0.37 | 0.71 |
| ServiceArea = Humanitarian (dummy) | 7.19 | 9.15 | 0.23 | 0.79 | 0.44 |
| ServiceArea = Military assistance (dummy) | 5.45 | 13.84 | 0.06 | 0.39 | 0.70 |
| ServiceArea = Association (dummy) | 34.16 | 13.61 | 0.35 | 2.51 | 0.02 * |

\* $p < 0.05$.

*RQ4: How do online accountability practices differ between domestic Ukrainian nonprofits and international non-governmental organizations (NGOs) operating in Ukraine?*

T-tests revealed that domestic nonprofits did not differ from NGOs in terms of overall OA practices, $t(89) = −1.115$, $p = $ n.s. Looking at individual dimensions, only governance showed significance. NGOs had much more robust governance accountability practices, compared to domestic nonprofits ($t(89) = −2.122$, $p < 0.05$; see Table 3).

**Table 3.** Weighted scores of OAP dimensions by nonprofit type.

| | Domestic NPOs (N = 59) | | NGOs (N = 32) | |
|---|---|---|---|---|
| | **Weighted M** | **Weighted SD** | **Weighted M** | **Weighted SD** |
| OAP | 34.7 | 14.7 | 38.5 | 17.5 |
| Accessibility | 8.2 | 3.1 | 7.3 | 3.3 |
| Engagement | 12.0 | 4.1 | 13.0 | 4.3 |
| Performance | 5.4 | 5.2 | 6.9 | 6.4 |
| Governance | 2.7 | 3.9 | 4.7 | 4.6 |
| Mission | 6.4 | 4.9 | 6.7 | 5.8 |

*RQ5: How do online accountability practices vary based on the service area (child welfare, education, humanitarian, etc.) of Ukrainian nonprofits?*

We ran the one-way ANOVA test to check for variance in online accountability practices among nonprofits within different service areas. In our analysis, we included the following service areas: child welfare, peace and reconciliation, humanitarian, animal welfare, military assistance, and association. Environmental and education nonprofits were excluded from the analysis because there was only one of each. We checked for homogeneity of variances. The significance value of the Levene statistic based on a comparison of

the medians was not significant, which means the requirement of homogeneity of variance was met, and the ANOVA test was considered to be robust.

We observed a statistically significant difference between the OAP means of the different service areas of the nonprofits ($F$(6,84) = 2.22, $p$ = 0.049), but the Tukey post hoc test comparisons between each pair of group means did not indicate any significant differences. The lack of significant results from the post hoc test could be attributed to insufficient sample sizes in each group, limiting the statistical power to find differences between the means.

We removed service area nonprofits that had been underrepresented in our sample, and focused on three groups that dominated the sample: child welfare, humanitarian, and other. Then, we ran another one-way ANOVA that revealed no statistically significant differences ($F$(2,88) = 2.86, $p$ = 0.063). Hence, we concluded that there was no variance in online accountability practices based on the service area of the nonprofits.

## 5. Discussion

This research project aimed to examine the use of online accountability practices by Ukrainian nonprofits and explore how these innovative tools can support strategic management. Specifically, we assessed the current landscape of website accountability adoption across the Ukrainian nonprofit sector. A key goal was determining what organizational characteristics are associated with greater utilization of online transparency and engagement features. The findings can provide insights into how nonprofits of different sizes, ages, and missions are leveraging technology for accountability. In addition, the results may point to management approaches and capacity issues influencing the integration of digital accountability practices. Overall, this research sought to understand the opportunities and challenges for Ukrainian nonprofits in employing online platforms for more effective transparency, stakeholder interaction, and organizational management. An enhanced adoption of web-based accountability tools has the potential to strengthen nonprofits' relationships with their communities and support their missions.

Our first finding was that Ukrainian nonprofit websites demonstrated wide variability in terms of online accountability practices, with some scoring quite low and others quite high on the weighted scale of the OAP Index. Looking across the five accountability dimensions, overall, Ukrainian nonprofits performed poorly in providing information related to governance and organizational performance on their websites, while achieving higher scores for engagement and accessibility features. This engagement emphasis aligns with previous research showing how NGOs prioritized attracting contributors rather than sharing details about performance (Uygur and Napier 2023). The accessibility scores showed the least variation among the sample, while performance scores varied the most. Notably, some websites provided no information at all about performance, governance, and mission, and were considered extremely difficult to access or navigate.

This project also focused on comparing online accountability management practices between domestic nonprofits and internationally based nonprofits (NGOs). While domestic nonprofits in Ukraine did not differ substantially from NGOs in overall online accountability practices, NGOs demonstrated notably more robust governance accountability compared to their domestic counterparts. This discrepancy in governance norms may be largely attributed to the differential organizational cultures and expectations placed on international NGOs versus locally grown Ukrainian civil society groups. As large global entities, international NGOs often import centralized policies and best practices from their institutional headquarters that reinforce strong governance structures. Ukrainian domestic nonprofits, by contrast, have typically emerged from a less structured civil society setting, leading to comparatively informal governance arrangements, as noted in previous scholarship characterizing Ukraine's nonprofit sector as a precarious "shadow third sector" (Krasynska 2015). Moreover, NGOs must satisfy strict accountability requirements imposed by international donors and supporters (Gandía 2011), which compels greater attention to transparent and legitimate governance mechanisms. Ukrainian domestic nonprofits enjoy

no such benefit, as studies have highlighted persistently low public trust and legitimacy hampering the local voluntary sector.

Ten nonprofits in our sample did not include their physical locations on their websites. Our analysis revealed that unknown locations of nonprofits are tied to significantly lower levels of online accountability. Not sharing a location may signal a general lack of strategic transparency and management. Another possible explanation is that these nonprofits have weaker community ties and limited external engagement. Nonprofits without a known location listed may not be strongly embedded within a geographic community, and this could translate to less public visibility and accountability pressures to innovate. It is also possible that these are organizations operating in challenging environments, especially considering the current climate in Ukraine. Nonprofits without a clear location may work in more insecure or risky environments where transparency could jeopardize operations. Security concerns could disincentivize full disclosure and use of digital tools. Further research on management approaches and innovative online practices is needed to understand how nonprofits lacking location transparency can strengthen their accountability given the constraints of their operating contexts.

Associations tended to have significantly higher online accountability practices compared to other nonprofit types in our sample. As member-based organizations, associations benefit from peer sharing of management innovations and accountability norms within their networks. Through informal collaborations and knowledge diffusion, associations are exposed to transparent best practices that spread rapidly across the field. This peer learning and accountability pressure motivates associations to adopt online innovations to demonstrate legitimacy. As pioneers and promoters of professional standards, associations can serve as role models and change agents for the broader Ukrainian nonprofit sector. Their professional orientation leads associations to proactively embrace proper management and governance practices seen as "best in class" reputation markers. By modeling cutting-edge accountability innovations, associations raise the bar and standards for the entire sector. Their competitive motivations, peer diffusion channels, and professionalism make associations potent catalysts for accelerating the adoption of online transparency tools as a norm.

Another finding of this study was the lack of variance in online accountability practices between Ukrainian nonprofits across different service domains. Rather than exhibiting divergence, online accountability practices appear largely uniform irrespective of whether organizations provide healthcare, education, social services, or other functions. This pattern points less toward peer standardization, whereby organizations model the behavior of compatriots in their field, and more toward indifference among influential sector stakeholders regarding differentiation in transparency norms. With the Ukrainian nonprofit sphere is still emerging, dominant players like government regulators and philanthropic funders may have yet to formulate distinct accountability expectations for diverse nonprofit subsectors. This stakeholder inattention to customizing accountability demands enables conformity born not of peer imitation but of the absence of pressures to tailor transparency efforts to individual organizational contexts. As the voluntary sector continues to evolve, stakeholders may sharpen their scrutiny and compel differentiation. For now, however, Ukrainian nonprofits of all varieties operate within a common normative horizon for online accountability practices, their convergence owing more to stakeholder inattention than peer emulation.

*Innovations, Challenges, and Best Practices*

This Special Issue focuses on nonprofit innovations, challenges, and best practices, and this study attempted to reveal both challenges and opportunities for innovation in online accountability practices among Ukrainian nonprofits. A key finding is the overall lack of performance and governance information disclosed on nonprofit websites. To enhance accountability, organizations should leverage platforms to share details on program outcomes, financial data, leadership structures, and policies. Interactive formats like annual

reports, searchable databases, and discussion forums can make this information engaging and accessible.

However, security and capacity issues pose barriers for nonprofits operating in high-risk environments. Organizations without a public location listing scored lower on online accountability, likely due to these challenges. Nonprofits confronting insecure conditions or limited resources may require creative transparency solutions like anonymized reporting and staggered disclosure policies. Partnerships with intermediary organizations could also help expand the technological capacity of small, grassroots groups through training programs and shared data platforms.

Associations seem well-positioned to champion wider transformation toward more accountable nonprofit management in Ukraine. Their capacities as networked incubators of innovation can drive faster propagation of online accountability practices. Associations have the scale, connections, competitiveness, and professional ethos needed to mainstream transparency innovations as a new standard for effective and ethical organizational leadership.

Ultimately, online accountability tools offer new avenues for nonprofits to demonstrate their impact and strengthen community trust. However, realizing this potential requires matching innovations to context. Organizations enjoying secure conditions can emulate global transparency norms by disclosing comprehensive, timely information on interactive websites. However, greater creativity is needed to tailor online accountability to the realities of groups working in restricted contexts. Ukraine's nonprofit sector would benefit from localized innovations that expand transparency while protecting vulnerable organizations.

## 6. Conclusions and Limitations

In conclusion, the examination of Ukraine's nonprofit sector points to opportunities for management innovation and strategic reorientation to strengthen nonprofit accountability. With low public trust hampering their legitimacy, Ukrainian nonprofits should pilot participatory initiatives that tangibly enhance people's well-being. As one study suggests, rather than pursuing broad democratization aims, focusing on projects that directly improve local conditions could help cultivate supporter relationships and organizational accountability (Gatskova and Gatskov 2016). Lasting accountability for Ukraine's nonprofits requires management approaches attuned to evolving public expectations and grassroots civic activism, along with political reforms addressing the climate of mistrust toward civil society. Navigating these complex, shifting dynamics will compel Ukrainian nonprofits toward creative experiments in accountable management and engagement.

This exploratory study has several limitations that should be noted. First, many service areas for nonprofits were underrepresented in our sample. With limited cases from fields like environmental and education nonprofits, we could not run a robust analysis to detect differences in online accountability between these different areas. Our sample also relied on a convenience approach rather than probability sampling from a complete nonprofit population list, since there is no publicly available database that fully enumerates all Ukrainian nonprofits. Additionally, we intended to include annual revenue in our model but found revenue information was only available for a fraction of groups, as no public data source compiles Ukrainian nonprofits' financials.

This study also used a quantitative content analysis to code and statistically analyze website content. While this enabled standardized comparisons, it provides a less rich interpretive analysis than qualitative methods could offer. By simplifying complex textual data into numeric scores, some deeper contextual meanings, implicit themes, subjective intents, and language nuances may be overlooked. Follow-up studies using qualitative approaches like interviews or inductive website analyses could strengthen these results by capturing undertones, motives, and explanations that quantitative techniques are less adept at revealing. A mixed-methods approach combining the breadth of quantitative content analysis with qualitative depth could provide a more comprehensive understanding of online accountability practices by Ukrainian nonprofits.

**Funding:** This research received no external funding.

**Institutional Review Board Statement:** Not applicable.

**Informed Consent Statement:** Not applicable.

**Data Availability Statement:** Data available upon request.

**Acknowledgments:** Two Oklahoma State University students, Taylor Perez and Shahariar Khan Nobel, contributed to this research project.

**Conflicts of Interest:** The author declares no conflict of interest.

## Appendix A

**Table A1.** Online Accountability Practices (OAP) Instrument.

| Accessibility Measures | | |
|---|---|---|
| **A1 *** | Does the website have navigation tools that appear on each page? | 3 = Yes; 0 = No |
| **A2 *** | Does the website have a search feature? | 3 = Yes; 0 = No |
| **A3 *** | Does the website have quick links? | 3 = Yes; 0 = No |
| **A4 *** | Does the website have a help feature? | 3 = Yes; 0 = No |
| **A5 *** | Does the website have a site map? | 3 = Yes; 0 = No |
| **A6 **** | What are the available language options? | 0 = One language; 1 = Two languages; 2 = Three languages; 3 = More than three languages. |
| **Engagement Measures** | | |
| **E1** | Does the site provide the date (month and year) of the **recent update** of a website? | 0 = No;<br>1 = the date is more than three months old;<br>2 = the date is more than one month and less than three months old;<br>3 = the date is one month or less old. |
| **E2** | Is there a subscribe option available for the **e-newsletter, listserv, magazine, etc.**? | 3 = Yes; 0 = No |
| **E3** | Does the site have a link for electronic **donations**? | 3 = Yes; 0 = No |
| **E4** | How many links to **social media** sites are there? | 0 = No social media sites listed;<br>1= 1–2 social media sites listed;<br>2= 3–4 social media sites listed;<br>3= 5 or more social media sites listed |
| **E5** | Does the site include options to **connect instantly** to the organization with one click (for example live chat or call now button)? | 3 = Yes; 0 = No |
| **E6** | Does the site provide an option to **share** the content? | 3 = Yes; 0 = No |
| **Performance Measures** | | |
| **P1** | Does the site post the organization's **annual report**? | 3 = Yes; 0 = No |
| **P2** | Does the site provide **financial statements** online? | 3 = Yes; 0 = No |
| **P3** | Does the site include **accreditation/honors/awards** information? | 3 = Yes; 0 = No |
| **Governance Measures** | | |
| **G1** | Does the site have the organization's **by-laws** available? | 3 = Yes; 0 = No |
| **G2** | Does the site list the names of the people who are on the **Board of Directors/Board of Trustees/Leadership Team**? | 3 = Yes; 0 = No |
| **G3** | Does the site publish **summary/minutes** from the Board of Directors meetings? | 3 = Yes; 0 = No |

Table A1. *Cont.*

| Mission Measures | | |
|---|---|---|
| **M1** | Does the site list the organization's **goals, strategic plan, or implementation plan**? | 3 = Yes; 0 = No |
| **M2** | Does the site offer an **employee directory** with contact information? (department contacts do not count) | 3 = Yes; 0 = No |
| **M3** | Does the site contain a **mission statement** for the organization? | 3 = Yes; 0 = No |
| **M4** | Does the site contain a statement of **values**? | 3 = Yes; 0 = No |

\* Based on Al-Qallaf and Ridha (2019), \*\* Added to this instrument.

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
