# Peer review of "Toward Greater Legitimacy: Online Accountability Practices of Ukrainian Nonprofits"

_admsci, doi:10.3390/admsci14010004_

Round 1

Reviewer 1 Report

Comments and Suggestions for Authors

Dear Authors,

Congratulations on your article.

The topic is very relevant, and non-profit organisations are increasingly important worldwide.

The article has a good structure.

I would suggest improving the methodology chapter. In my opinion, the use of different tests for each question is confusing. Perhaps a summary table or other detail should be included in the explanation.

They could also indicate which software was used.

In the references, they need to correct the use of block letters in the following reference:

USAID. 2021. "2020 CIVIL SOCIETY ORGANISATION SUSTAINABILITY INDEX: UKRAINE." https://rpr.org.ua/wp-602 content/uploads/2021/11/CSOSI-Ukraine-2020.pdf.

Best of luck with your article.

Best regards,

Author Response

Dear Reviewer,

We appreciate the feedback from you and the opportunity to make further revisions on our manuscript for the Administrative Sciences journal. We have implemented all of the suggestions and believe that the manuscript is much strengthened as a result. Below, we list the feedback we received, and detail how we have responded to each concern in our revision of the manuscript (highlighted in yellow).

Reviewer 1: I would suggest improving the methodology chapter. In my opinion, the use of different tests for each question is confusing. Perhaps a summary table or other detail should be included in the explanation. They could also indicate which software was used.

Response: Thank you for the suggestion to improve the clarity of the methodology chapter regarding the statistical tests used for each research question. We used SPSS version 26 for all data analyses. I included this information in the text. Different statistical tests were used because they were considered most appropriate to answer the posed research questions. For example, to answer RQ3, we ran a multiple regression. Our DV was continuous data (OAP instrument), and IVs were mostly dummy variables (with the exception of age, which was a continuous variable). RQ4 asked about differences in OAP practices between two groups, and T-test is the most appropriate statistical test to determine that. RQ5 asked about differences between multiple groups, so we couldn’t run another t-test. We had to run a one-way ANOVA. Please let me know if you have any other recommendations for improving transparency on test selection decisions or highlighting their appropriateness.

Reviewer 1: In the references, they need to correct the use of block letters in the following reference: USAID. 2021. "2020 CIVIL SOCIETY ORGANISATION SUSTAINABILITY INDEX: UKRAINE." https://rpr.org.ua/wp-602 content/uploads/2021/11/CSOSI-Ukraine-2020.pdf.

Response: This has been now corrected.

Reviewer 2 Report

Comments and Suggestions for Authors

Review for Toward Greater Legitimacy: Online Accountability Practices of Ukrainian Nonprofits

The authors present an interesting idea, made more so by the influx of international NGOs into Ukraine following Russia’s invasion.  Accountability is crucial for non-profits and NGOs if they want to build trust and be sustainable.

The approach was interesting and made sense given the information readily available. The article was well written and pretty easy to understand and follow, even if someone is not familiar with NGOs and accountability. The literature review was well done for the most part and well organized to take people through the aspects and important issues of this article.

However, there are some concerns.

1)     The references for the overall view of the nonprofit sector in Ukraine are older articles, there is nothing younger than five years old.  This may be because there is nothing more recent, but I wonder how much has changed in the last five years, given everything that has happened in the world.

2)     The researchers are using quantitative analysis, but have research questions.  Research questions are better suited for qualitative data, it would have been better if they had hypotheses instead.  It would have made the analysis more straight forward.  It is difficult to really answer the how RQ with the data provided. 

3)     When discussing the coding method, they refer to location as either Kyiv, regional, international or unknown.  Does Kyiv mean a national NPO? That is not clear.  Furthermore, while they spelled Kyiv using the form traditionally used in Ukraine in line 278 (page 6), they spell it using the western incorrect pronunciation  (Kiev) twice on page 8, there needs to be consistency,

4)     The discussion on the other/unable to determine location was interesting, but it would be more interesting if we knew how many of the organizations in the sample fell into that category and a clearer idea of why. 

5)     For RQ5, the authors mention using a category of other without any expansion on what the other category consists of and why an organization would be assigned to that category. 

6)     The authors discuss that they are using both nonprofit organizations and NGOs.  Did they assume that NGOs are international and NPOs are domestic only?  Some better clarification of this would be helpful. Perhaps a clear definition of what they mean by non- profit and NGO.  Domestic NGOs are common in the CEE region.

I do agree with the authors that the quantitative data analysis means that they cannot capture some of the underlying interpretive analysis that would have helped answer some of those ‘how’ questions better.  Given the current situation in Ukraine, I understand why they did not go the qualitative route, but encourage them to do so when the situation makes it possible. 

Comments on the Quality of English Language

only one minor issue the two spelling of Kyiv or Kiev, I recommened they use Kyiv consitently.

Author Response

Dear Reviewer,

We appreciate the feedback from you and the opportunity to make further revisions on our manuscript for the Administrative Sciences journal. We have implemented all of the suggestions and believe that the manuscript is much strengthened as a result. Below, we list the feedback we received, and detail how we have responded to each concern in our revision of the manuscript (highlighted in yellow).

Reviewer 2:   The references for the overall view of the nonprofit sector in Ukraine are older articles, there is nothing younger than five years old.  This may be because there is nothing more recent, but I wonder how much has changed in the last five years, given everything that has happened in the world.

Response: We have added a new citation to the literature review for the article that was published in 2023. This article focused on the informal, value-driven nature of civil society in Ukraine that manifests itself strongly during historical junctures like Euromaidan and the current war. The manuscript now reads: “Ukrainian understandings of civil society tend to emphasize informal action, values, responsibility, and sense of community rather than just membership in formal organizations (Martin and Zarembo 2023). Not surprisingly, events like Euromaidan and the current Russian invasion have triggered massive grassroots civic mobilization (Martin and Zarembo 2023). The concept of "sense of community" helps explain the dormant yet powerful nature of Ukrainian civil society during crises. The strong values, emotional connections, and responsibility Ukrainians feel towards their nation and community drive heightened civic activism when threats emerge (Martin and Zarembo 2023).” This newly added article does not cite any recent studies focusing on Ukrainian nonprofits. There are two recent public opinion polls and one Wall Street article.

Reviewer 2: The researchers are using quantitative analysis, but have research questions.  Research questions are better suited for qualitative data, it would have been better if they had hypotheses instead.  It would have made the analysis more straight forward.  It is difficult to really answer the how RQ with the data provided. 

Response: Thank you for raising this issue regarding the use of research questions rather than hypotheses in the context of a quantitative analysis. I appreciate you noting that hypotheses are more common and potentially better suited for such data.

The editor recommended retaining the research question framing given this is positioned as an exploratory study without specific predictions going into data collection and analysis. The editor clarified that for this initial investigation exploring accountability adoption factors, the research question approach still seems appropriate as it offers more flexibility for the data to guide findings.

Reviewer 2:   When discussing the coding method, they refer to location as either Kyiv, regional, international or unknown.  Does Kyiv mean a national NPO? That is not clear.  Furthermore, while they spelled Kyiv using the form traditionally used in Ukraine in line 278 (page 6), they spell it using the western incorrect pronunciation  (Kiev) twice on page 8, there needs to be consistency.

Response: Thank you for noting the confusing terminology regarding nonprofit location coding and the inconsistent spelling of the Ukrainian capital. To clarify, nonprofits headquartered in Kyiv were coded as "Kyiv" while those located elsewhere in Ukraine were designated as "regional." Only nonprofits based outside Ukraine altogether were considered "international." Also, we have now thoroughly reviewed the manuscript to correct the capital's spelling to the Ukrainian language standard of "Kyiv" throughout for consistency.

Reviewer 2:     The discussion on the other/unable to determine location was interesting, but it would be more interesting if we knew how many of the organizations in the sample fell into that category and a clearer idea of why. 

Response: Thank you for noting the lack of specifics around the number and potential reasons behind nonprofits not listing location information. To address this, we have added that 10 nonprofits (around 11% of the sample) fell into the "unknown location" coding category. As for why locations may be unspecified, we unfortunately do not have further data beyond conjecture. Our speculation is that absent location details may signal weaker transparency practices and strategic management overall. But without additional data on the characteristics or rationale from those 10 nonprofits, we cannot substantiate reasons for the missing location information. Please let us know if any other supplemental details would help contextualize this subset of nonprofits lacking location transparency.

Reviewer 2:   For RQ5, the authors mention using a category of other without any expansion on what the other category consists of and why an organization would be assigned to that category. 

Response: Thank you for catching the confusing inclusion of an "Other" category for service area coding that was never actually utilized. You raise a fair point that mentioning this unused code in the scheme without explanation or examples was problematic. Upon double checking the final service area designations during analysis, we verified that no nonprofits were actually assigned this "Other" code. Therefore, we have removed the category from the coding guide altogether to eliminate confusion since it did not end up containing any observations. Please let us know if any other details on the service area coding scheme would help clarify the final set of categories used based on where nonprofits indicated targeting their activities and benefiting communities.

Reviewer 2: The authors discuss that they are using both nonprofit organizations and NGOs.  Did they assume that NGOs are international and NPOs are domestic only?  Some better clarification of this would be helpful. Perhaps a clear definition of what they mean by non- profit and NGO.  Domestic NGOs are common in the CEE region.

Response: Thank you for raising the need to clarify terminology and assumptions around nonprofits vs NGOs. You are correct that we equated domestic groups as NPOs and international groups as NGOs, which is an imperfect system. There are certainly domestic NGOs active in Ukraine and NPOs can have international affiliations. However, drawing stark lines was necessary for comparison purposes within project constraints. Going forward, explicitly defining these categories as domestic headquarters (NPOs) vs international headquarters (NGOs), rather than claiming they fully capture nonprofit legal forms, will help transparently convey the quantitative shortcuts made. We aim to acknowledge the messy reality within Ukraine's civil society landscape while defending our binary approach as the feasible delineation mechanism for this first step exploration. Please let us know if you have any other recommendations for appropriately scoping language around the NPO/NGO divide as more complex than a true legal form-based distinction.

Reviewer 2: Only one minor issue the two spelling of Kyiv or Kiev, I recommened they use Kyiv consitently.

Response: This has been corrected.